# Haematological Drugs Affecting Lipid Metabolism and Vascular Health

**DOI:** 10.3390/biomedicines10081935

**Published:** 2022-08-10

**Authors:** Antonio Parrella, Arcangelo Iannuzzi, Mario Annunziata, Giuseppe Covetti, Raimondo Cavallaro, Emilio Aliberti, Elena Tortori, Gabriella Iannuzzo

**Affiliations:** 1Department of Medicine and Medical Specialties, A. Cardarelli Hospital, 80131 Naples, Italy; 2Hematology Unit, A. Cardarelli Hospital, 80131 Naples, Italy; 3North Tees University Hospital, Stockton-on-Tees TS19 8PE, UK; 4Pharmacy Unit, Ospedale del Mare, 80147 Naples, Italy; 5Department of Clinical Medicine and Surgery, Federico II University, 80131 Naples, Italy

**Keywords:** haematological drugs, side effects, lipid metabolism, hypertriglyceridaemia, hypercholesterolemia, atherosclerosis, vasculopathy

## Abstract

Many drugs affect lipid metabolism and have side effects which promote atherosclerosis. The prevalence of cancer-therapy-related cardiovascular (CV) disease is increasing due to development of new drugs and improved survival of patients: cardio-oncology is a new field of interest and research. Moreover, drugs used in transplanted patients frequently have metabolic implications. Increasingly, internists, lipidologists, and angiologists are being consulted by haematologists for side effects on metabolism (especially lipid metabolism) and arterial circulation caused by drugs used in haematology. The purpose of this article is to review the main drugs used in haematology with side effects on lipid metabolism and atherosclerosis, detailing their mechanisms of action and suggesting the most effective therapies.

## 1. Introduction

Increased LDL-cholesterol (LDL-C) concentrations are causally related to atherosclerotic cardiovascular disease (ASCVD) [1]. However, many patients with normal LDL-C levels (genetically determined or after cholesterol lowering therapy) still have a residual CV risk. Among the principal metabolic lipid-related CV risk factors implicated in this residual risk, hypertriglyceridaemia, atherogenic lipoprotein subfractions, reduced concentrations or altered function of HDL and elevated levels of Lp_a_ are sure to play a significant role [1,2,3,4]. Many dyslipidaemias are of genetic origin (primary dyslipidaemias), but others are secondary to other conditions such as: diabetes mellitus, hypothyroidism, chronic kidney disease, cholestasis, and drugs.

It is well known that many drugs have the ability to influence the lipid profile. In some cases, this is an effect of the drug class; in other cases, different agents belonging to the same class can have significantly different effects on lipid levels [5,6].

This aspect is of considerable importance when initiating specific treatments for patients, especially when the CV risk is high or very high. For example, some antihypertensives can increase the plasma levels of triglycerides (TG), notably thiazide diuretics and β-blockers [7]. The triglyceride-increasing effects of the latter vary, with greater effects seen with propranolol, metoprolol, and atenolol, and more generally for nonselective agents [8]. Antipsychotics and antiretroviral drugs are also known to be associated to dyslipidaemias. Some second-generation antipsychotic medications such as clozapine, olanzapine, risperidone, and quetiapine can be associated with hypertriglyceridaemia; however, this effect has not been seen with aripiprazole or ziprasidone [9]. Antiretroviral therapy (ART) in HIV patients has been associated with metabolic changes, particularly an increased prevalence of insulin resistance [10]. Older protease inhibitors (PIs), such as saquinavir, indinavir, and ritonavir, sometimes lead to dyslipidaemia. Newer PI therapies, such as darunavir/ritonavir or atazanavir/ritonavir can still cause mild dyslipidaemia, much less dramatic than older PIs. Some older nucleoside reverse-transcriptase inhibitors (NRTIs), such as stavudine and zidovudine, sometimes cause dyslipidaemia, particularly in combination with PIs. Efavirenz is the only non-NRTI (NNRTI) that could slightly raise LDL-C. Other new ARTs (tenofovir, nevirapine, rilpivirine, raltegravir, dolutegravir, and bictegravir) have no significant effects on lipids or have a lipid-lowering effect [11]. Many other drugs have been implicated in the pathogenesis of secondary dyslipidaemias.

In recent years, there has been a particular focus on the correlation between haematological diseases, particularly neoplastic diseases, and lipid metabolism and atherosclerosis. Recent research has demonstrated that this correlation is often due to intrinsic mechanisms of the disease and its degree of severity. For example, changes in lipid metabolism have been found in chronic lymphocytic leukaemia (CML) with increased levels of total cholesterol and low-density lipoproteins (LDL) and decreased levels of high-density lipoproteins (HDL), especially in advanced stages of the disease. Similar results have been found in patients with non-Hodgkin lymphoma [12]. In other cases, changes in lipid metabolism are direct consequences of drugs.

In a retrospective study, the prevalence, clinical characteristics, and risk of severe pancreatitis were analysed in a large number of children with severe hypertriglyceridaemia (>2000 mg/dL) [13]. It showed that around 28% of these children were diagnosed with ALL and treated with L-asparaginase and high doses of steroids. More than half of hypertriglyceridemic patients analysed in that study subsequently developed complications including at least one case of pancreatitis and the development of insulin-dependent diabetes mellitus. Therefore, it is important to diagnose drug-induced hyperlipidaemia early to avoid CV complications or acute pancreatitis.

This review aims to evaluate the effect of commonly used drugs in haematology on lipid metabolism and vascular health and tries to clarify the underlying mechanisms involved.

## 2. Tyrosine Kinase Inhibitors

The treatment of chronic myeloid leukaemia (CML) has been revolutionised with the introduction of oral tyrosine kinase inhibitors (TKIs). It should be remembered that following approval of TKIs for the treatment of CML, the overall survival of these patients improved to 70% [14]. Imatinib (a first-generation inhibitor) can reduce serum levels of proatherogenic LDL and can normalise plasma lipid levels in hypercholesterolemic and hypertriglyceridemic patients [15]. However, imatinib therapy sometimes has adverse events unacceptable to the patient such as oedema (periorbital and peripheral), musculoskeletal pain, muscle cramps, diarrhoea, etc.

Nilotinib is a second-generation TKI licensed for first and subsequent line treatment of CML, with greater potency and affinity for the BCR-ABL1 oncoprotein compared to imatinib. In the ENESTnd study (a >10 years follow-up study of nilotinib versus imatinib in CML patients) nilotinib demonstrated lower rates of disease progression and disease-related death as well as a higher cumulative molecular response rate [16]. In the same study, CV events were reported in 16.5% of patients taking nilotinib 300 mg twice daily, 23.5% of patients taking nilotinib 400 mg twice daily, and 3.6% of patients on imatinib therapy. Patients > 60 years had a higher incidence of CV events: 34.5%, 34%, and 8.5%, respectively. In addition to peripheral arterial disease (PAD) other serious occlusive arterial diseases, such as coronary artery disease and cerebrovascular ischaemia, have been reported in various clinical studies, including retrospective and case studies [17]. Moreover, Bondon et al. demonstrated that the incidence of PAD was higher above 60 years of age, and the mean onset of disease was after 24 months of therapy. In the rare cases of PAD in patients younger than 60 years old, the time to onset after nilotinib therapy appeared sooner [18]. In a study by Kim et al., 129 patients with CML were screened for PAD during treatment with TKIs [19]. A pathological ankle brachial index (ABI) < 0.9 was documented in around 6% of patients treated with imatinib as first line therapy, compared to 26% of patients who were treated with nilotinib as first line therapy and 35% of patients if nilotinib was given as second line therapy. Tefferi et al. described a case of death due to severe PAD in a patient with CML treated with nilotinib, and other cases of patients in which treatment was associated with rapidly progressing intracranial and extracranial atherosclerosis, with subsequent ischaemic stroke [20].

The mechanism by which nilotinib exerts its proatherogenic effect is not clear and is likely multifactorial. In a mouse model of atherosclerosis, nilotinib promoted aortic wall atherosclerosis in ApoE-/- mice, whereas imatinib showed no proatherogenic effect. In this study it was shown that nilotinib inhibits a series of kinases involved in the “repair” of stress-induced vascular damage (JAK1, TEK) which are not normally inhibited at a therapeutic dose of imatinib. Moreover, nilotinib was found to exert a proatherogenic effect acting directly on human endothelial cells through the upregulation of adhesion proteins ICAM-1, E-selectin, and VCAM-1 [21]. In other words, this proatherogenic effect could contribute to creating vessel stenosis, whilst the antiangiogenic effects would stop defensive mechanisms from recanalising and repairing vessels once stenosis is established. Another contributing factor appears to be this inhibitor’s ability to create vasospasm that can trigger the entire pathophysiological mechanism [22].

Hypercholesterolaemia has been described as a side effect during treatment with nilotinib, which is closely correlated with the development of peripheral arteriopathy. In particular, the use of nilotinib is associated with an increase in total cholesterol, with increases in both HDL and LDL. The risk factors for hypercholesterolemia during nilotinib therapy are older age, duration of treatment, and pre-existing metabolic risk [23]. In the study by Rea et al., at baseline, plasma cholesterol concentration was 180 ± 38 mg/dL (mean ± SD); after three months of therapy with nilotinib, cholesterol levels had risen by a mean of 45 mg/dL, with a mean plasma cholesterol concentration of 224 ± 47 mg/dL [23]. Interestingly, triglyceride concentrations decreased by a mean of −35 mg/dL between baseline and three months. An Italian study evaluated LDL levels along with arterial occlusive events in 369 CML patients treated with nilotinib; the total cholesterol and LDL-cholesterol concentrations increased during nilotinib therapy: the total cholesterol concentrations were 185 mg/dL (median), (76–305 range), at baseline and raised to 207 mg/dL (median), (99–305 range), *p* < 0.001, after 3 months. HDL and triglycerides did not show significant concentration changes. Patients with hypercholesterolemia 3 months after treatment (total cholesterol >200 mg/dL and LDL > 70 mg/dL) showed a significantly higher incidence of arterial vasculopathy (21.9% vs. 6.2%, *p* < 0.01), but, surprisingly, only a small proportion of hypercholesterolemic patients (29.5%) were treated with statins [24].

The mechanism by which the second-generation inhibitors induce hyperlipoproteinemia is not clear. One hypothesis is that this increase is a secondary consequence of another side effect of this drug, which is insulin resistance and hyperinsulinemia, as well as a decreased synthesis of lipoprotein lipase (LPL). A small clinical trial showed that nilotinib significantly increased PCSK9 plasma concentration after 3 months of therapy and suggested a possible role of this in nilotinib-induced hypercholesterolemia [25]. Other studies have hypothesised that nilotinib is possibly directly toxic at the level of adipose tissue [26]. The effect on lipid metabolism along with a direct effect on the endothelium could explain the increase in CV risk seen in this patient group.

Another tyrosine kinase inhibitor which is implicated in vascular damage is ponatinib. Ponatinib is a third-generation TKI and is an excellent drug in individuals who show a poor response to therapy in Philadelphia+ ALL and in CML. The PACE trial reported that the incidence of arterial thrombotic events with ponatinib was around 9% for those patients treated for 11 months and around 17% when treated for 24 months [27]. However, it should be noted that most of the patients in that study had previously been treated with nilotinib. A more recent study demonstrated a smaller incidence of CV events compared to the PACE trial, but an accurate comparison could not be made due to various confounding factors [28]. In any case, Caocci et al. recently showed that the incidence of vascular events in patients treated with ponatinib are greater if upon commencing treatment, the triglyceride level is >200 mg/dL, and if the total cholesterol is >200 mg/dL and LDL is >70 mg/dL after three months from the start of ponatinib. For this reason, the authors suggested initiating statin therapy before commencing treatment with ponatinib [29]. The mechanism by which ponatinib induces thrombosis is not fully understood. Animal studies in mice have demonstrated that ponatinib therapy alters both vessel wall and platelet function [30].

Hypercholesterolaemia is only one pre-existing risk factor which can favour the occurrence of vascular events with tyrosine kinase inhibitors. Other risk factors are hypertension, smoking, diabetes mellitus, and age. CV events occur more frequently in patients with high or intermediate baseline Framingham risk than in the low-risk category. In these patients, reducing the dose by around half can reduce the incidence of CV events by around 33% [31].

An echo colour doppler is one of the best methods to noninvasively evaluate subclinical atherosclerosis and to follow up atherosclerotic lesions, especially in subjects with dyslipidaemia [32].

For the aforementioned reasons, in patients with CML who are candidates for treatment with nilotinib or ponatinib, it is advisable to perform metabolic and vascular screening to evaluate CV risk factors. Our suggestion would be to investigate for the presence of hypertension, diabetes mellitus, dyslipidaemia, hypothyroidism, smoking, abnormal renal function, and acquire measurements for ankle–brachial pressure indices, duplex ultrasound of the carotid arteries, and peripheral vessels of the lower limbs, and finally echocardiographic measurement of the ejection fraction.

By way of example, Table 1 shows anthropometric and CV baseline data for 35 patients undergoing treatment with nilotinib for CML in our Division of Haematology at the A. Cardarelli Hospital in Naples (Italy); Figure 1 shows the frequency (in our patient cohort) of the presence of carotid and femoral plaques prior to initiating second- and third-generation tyrosine kinase inhibitors.

## 3. Janus Associated Kinase (JAK) Inhibitors

Ruxolitinib is a JAK1/2 inhibitor used for the treatment of myelofibrosis and polycythaemia vera intolerant or resistant to hydroxyurea due to its efficacy in the symptomatic reduction and treatment of splenomegaly typical of these diseases, and is associated with improved survival compared to placebo. In addition, there are a number of trials underway to evaluate the safety and efficacy of ruxolitinib for cases of graft-versus-host disease (GvHD) in allograft recipients who are refractory to treatment with corticosteroids [33].

Ruxolitinib improves metabolism in patients with myelofibrosis because this disease is characterised by abnormally low cholesterol concentrations, which have been associated with shortened survival. As reported by Mesa et al., levels of total cholesterol increased in patients treated with ruxolitinib compared to pretreatment levels of the same group. Patients receiving ruxolitinib experienced a mean 35.8% increase in total cholesterol (mean increase: 38.0 mg/dL); however, for the majority of patients, the total cholesterol concentration did not exceed 240 mg/dL. Given that hypocholesterolaemia is associated with poorer prognosis in these patients, the improvements in total cholesterol levels after ruxolitinib therapy could represent disease-modifying effects that contribute to a better prognosis than placebo [34].

The summary of product characteristics of ruxolitinib mentions hypertriglyceridaemia as a common although “mild” adverse effect. Sometimes hypertriglyceridaemia during treatment with ruxolitinib (especially if associated with sirolimus) could be severe and life-threatening. Watson et al. reported a case of hypertriglyceridaemia in a 50-year-old patient treated with a combination of ruxolitinib and sirolimus for chronic GvHD. TG levels were between 300 and 400 mg/dL over the last several years but peaked to 2983 mg/dL after sirolimus and ruxolitinib therapy. TG levels only returned to basal levels once JAK1 inhibitors had been suspended [35]. A similar case was reported by Bauters et al. in a patient who already had raised TG levels due to sirolimus, but these values were further elevated once ruxolitinib was added (to around 10,000 mg/dL), resulting in pancreatitis. The effects of treatment with this drug on CV outcomes are not known, and for this reason both authors suggest monitoring the lipid profile at initiation of treatment and after 8–12 weeks, especially in patients where it is used alongside sirolimus [36].

The mechanism behind this dyslipidaemia and weight gain seems to correlate with the dysregulation of leptin levels at a hypothalamic level mediated by JAK2/STAT3, a phenomenon which can cause hyperlipidaemia, dysphagia, and changes in glucose metabolism. In fact, ruxolitinib seems to interrupt postprandial leptin signalling pathways, which can cause hyperphagia, which contributes to the increases in weight seen in the majority of patients treated [37].

## 4. PEG-Asparaginase

Asparaginase is a key drug in most protocols used for the treatment of ALL, above all during the induction phase and intensification phases of the treatment. Survival of both children and adults affected by this disease has improved since the introduction of this drug in chemotherapy protocols [38]. Among numerous side effects are abnormalities affecting lipid metabolism, predominantly hypertriglyceridaemia [39]. The exact mechanism causing lipid abnormalities is not fully understood and still needs clarification. It has been suggested that it could be related to an increased endogenous hepatic synthesis of VLDL. Another suggested mechanism is a reduced enzymatic activity of LPL, a key enzyme in the removal of triglyceride-rich lipoproteins from plasma, reducing clearance and consequently causing hypertriglyceridaemia [40]. This latter aspect is more evident when used in combination with corticosteroids. In fact, systemic steroid therapy normally brings about increased concentrations of triglyceride-rich VLDL, but this is balanced by an increased activity of LPL induced by the steroids. When steroids are used in combination with L-asparaginase, this latter effect is lacking, and hypertriglyceridaemia is a typical finding in some patients treated with both drugs. The way in which L-asparaginase decreases the catalytic activity of LPL is not clear but appears to be due to the inhibition of synthesis [41]. A study by Tong et al. showed that the combined use of L-asparaginase and steroids was associated with changes in lipid metabolism in more than 50% of cases. That study also demonstrated that the incidence of hypertriglyceridaemia and hypercholesterolaemia was higher in the cohort of patients treated with the pegylated-asparaginase (PEG-ASP) than with the *Erwinia* formulation. PEG-ASP-induced hypertriglyceridaemia did not, however, correlate with an increased incidence of severe clinical events such as acute pancreatitis and/or arterial thrombosis. In that study the median triglyceride level was 71 mg/dL and the median cholesterol level was 143 mg/dL at baseline. After PEG-ASP therapy, the concentration of TG rose to 280 mg/dL (*p* < 0.001) and remained stable at high values throughout the course of therapy. Cholesterol concentrations after 9 weeks of therapy peaked to 240 mg/dL (*p* < 0.001) and remained stable thereafter. The triglyceride and cholesterol levels normalised in all patients 8 weeks after the end of the PEG-ASP therapy [42]. Finch et al. found the same results. However, they observed that the treatment with PEG-ASP carried significant increases in TG compared to L-asparaginase, when used in combination with identical doses of dexamethasone. Even so, in that case, there was not an increased incidence of CV or gastroenterological side effects [43]. In newly diagnosed children with ALL, it was observed a 67% incidence of hypertriglyceridaemia, 19% with concentrations greater than 1000 mg/dL [39].

The increase in TG therefore appears to be rarely of any clinical importance and is transitory and so does not always require suspension of therapy. In any case, other studies have reported satisfactory results with the use of omega-3, fibrates, or temporary suspension of dexamethasone therapy. Some cases of severe asparaginase-induced hypertriglyceridaemia have been successfully managed with plasmapheresis, which was found to be a safe and effective method for treating hypertriglyceridaemia and preventing related complications [44].

Recently, it has been proposed that the increase in triglycerides to the particularly high levels which is observed in some individuals treated with PEG-ASP could be due, at least in part, to genetic predisposition. In particular, it was seen that the presence of a rare missense variant (c.11 G > A-p(arg4Gin)) in the ApoCIII gene along with four other common single nucleotide polymorphisms (SNPs; c.*40 C > G in APOCIII e c.*158 T > C; c.162-43 G > A; c.-3 A > G in ApoA5) were associated with a case of severe hypertriglyceridaemia in an adult patient with ALL after two cycles of therapy with PEG-ASP and steroids [45].

## 5. Calcineurin Inhibitors

It is generally accepted that therapy with calcineurin inhibitors (CNIs) is associated with hyperlipidaemia. Cyclosporine and tacrolimus are immunosuppressive agents that play a pivotal role in patients receiving organ transplantation. These drugs are also used in some haematological diseases such as aplastic anaemia, haemophagocytic syndrome and Castleman disease [46]. Particular attention has been given in previous years to the use of cyclosporine and tacrolimus following haematopoietic stem cell transplantation. In fact, GvHD represents one of the most common causes of mortality, not related to disease relapse, following transplantation.

Introduced in the 1980s, these immunosuppressive agents, which prevent the activation of T-cells by inhibiting calcineurin, have dramatically improved survival rates of the allograft recipients. The first studies that reported very beneficial results of regimes using CNIs date back to 1986, with a significant reduction of GvHD and improved survival compared to the use of only one of the two agents in prophylaxis protocols [47]. Two multicentre studies, randomised and prospective, conducted in the mid-1990s demonstrated a lower incidence of GvHD with tacrolimus plus methotrexate compared to cyclosporine plus methotrexate, although overall survival was not significantly different [48]. A recent study has confirmed the ability of cyclosporine in both acute and chronic GvHD prophylaxis; the addition of mycophenolate mofetil did not reduce the incidence or severity of this complication in patients undergoing allogenic transplant for acute myeloid leukaemia [49].

CNIs have important and serious side effects: nephrotoxicity, hypertension, diabetes mellitus, and dyslipidaemia. Different metabolic abnormalities are associated with the use of CNIs, including glucose intolerance, osteoporosis, and increased levels of total cholesterol (TC), LDL and apolipoprotein B100 (apo B-100). The effects on HDL are less relevant, although some studies have shown increases in this lipoprotein [6]. Hyperlipidaemia can occur in up to 60% of patients post-transplant [50]. This is due to multiple factors including post-transplant obesity, side effects of other drugs (including steroids and other immunosuppressors), and diabetes. The effects on lipid metabolism are much more evident with cyclosporine compared to tacrolimus, which seems to have fewer effects on TC and LDL. However, it is also important to mention that the dose of corticosteroid used is higher in patients treated with cyclosporine than with tacrolimus and could introduce a bias in a comparison between the two drugs in affecting lipid metabolism. One prospective randomised study compared a regime based on tacrolimus to a regime based on cyclosporine in patients following heart transplantation. After 12 months of therapy, TC, LDL, and TG were significantly higher in the cyclosporine group compared to the tacrolimus group. In particular, LDL rose from 115 ± 36 mg/dL (baseline, mean ± SD) to 131 ± 38 (cyclosporine) and to 116 ± 28 (tacrolimus); TG rose from 156 ± 109 to 218 ± 117 (cyclosporine) and to 187 ± 128 (tacrolimus) [51]. Zimmerman et al. investigated changes in lipid metabolism under CNI (78 patients) and mTOR (14 patients) immunosuppressive regimens after liver transplantation. LDL-C (mg/dl) was 118.2 ± 36.7 (mean ± SD) at baseline, 115 ± 34 after CNI-only containing regimen and 139 ± 46 (*p* < 0.047) after mTOR-containing regimen. TG (mg/dl) values were, respectively, 177 ± 134, 160 ± 129, and 258 ± 122 (*p* < 0.001) [52]. On the other hand, there are conflicting data regarding the effect on small dense LDL (sdLDL), whose concentration has been associated with an increased risk of CV disease in the general population [53]. The impact of cyclosporine and tacrolimus on lipid levels seems to be dose-dependent and baseline lipid levels play a significant role in TC and LDL concentrations after therapy with these drugs. Some patients develop moderate to severe hypertriglyceridaemia after using CNIs, while they rarely show severe hypercholesterolemia. The mechanism by which cyclosporine causes hyperlipidaemia is not completely clear. Different studies have demonstrated that cyclosporine increase Ser^552^ phosphorylation in adipose tissue by hormone sensitive lipase and contribute to the development of insulin resistance [54]. Cyclosporine has a dose-dependent lipid-increasing effect; the drug, as part of its metabolism, inhibits the sterol 27-hydroxylase (CYP27A1), reduces the activity of sterol 27-hydroxylase, and in this way, slows the catabolism of cholesterol through bile acids; it promotes the synthesis of cholesterol due to a lack of inhibition from hydroxymethylglutaryl-CoA reductase (HMG-CoA reductase) by 27-hydroxycholesterol [55]. It also binds to the LDL receptor causing an increase in LDL cholesterol and reducing clearance of VLDL [56,57]. Moreover, through the inhibition of LPL, it induces hypertriglyceridaemia [58,59]. Lastly, cyclosporine induces an increase in apoCIII (endogenous inhibitor of LPL) that could result in hypertriglyceridaemia and an increased concentration of small dense LDL [60].

Tacrolimus shares many of the effects on lipid balance with cyclosporine, and the mechanisms of action are also often the same. The main mechanism linking tacrolimus with hyperlipidaemia (especially hypertriglyceridaemia) and diabetes mellitus is increased insulin resistance [61]. Another mechanism in tacrolimus-related dyslipidaemia was proposed by Zhang et al., who observed that tacrolimus could induce TG accumulation in hepatocytes and dyslipidaemia by downregulating a circRNA (circFANS), stimulating a microRNA (miR-33A), and dysregulating SREBPs (sterol regulatory element-binding protein) [62]. Moreover, tacrolimus potentiates the effect of glucolipotoxicity, decreasing the Akt phosphorylation and reducing β-cell proliferation [63].

The serum levels of drugs need to be monitored during treatment, as increased levels are associated with adverse effects. Furthermore, if appropriate, switching from cyclosporine to tacrolimus can be considered if hyperlipidaemia occurs, as many studies have shown [64]. Although patients could benefit from treatment with HMG-CoA reductase inhibitors, it has been shown that the concomitant use of cyclosporine with HMG-CoA reductase inhibitors increases the risk of myopathy and rhabdomyolysis due to a potential drug–drug interaction caused by the inhibition of the CYP3A-mediated metabolism of simvastatin and the inhibition by cyclosporine of the hepatic absorption of simvastatin mediated by organic ion transporter proteins (OATP1B1) [65]. Many statins (atorva, lova, simva, etc.) compete with CNIs for the same hepatic metabolising enzymes (cytochrome P450 3A4 isoenzyme) and can increase blood concentration of the CNIs; since pravastatin and fluvastatin are not metabolised in a significant way by CYP enzymes, they could be a favourable choice in this group of patients due to a decreased risk of drug–drug interactions [66].

## 6. Mammalian Target of Rapamycin Inhibitors

The mammalian target of rapamycin (mTOR) is a downstream target of many signalling pathways. The most famous is the PI3k/Akt/mTOR signalling pathway, which has a central role in cell growth, proliferation, differentiation, and survival, in protein synthesis, and in the metabolism of glucose and lipids. The mTOR pathway regulates lymphoid and myeloid development and function. The aberrant regulation or hyperactivation of mTOR is a distinctive sign of many tumours, including haematological malignancies. Various pharmaceutical agents which inhibit the mammalian target of rapamycin have been developed, since they induce the apoptosis of neoplastic cells, arrest of the cell cycle, and inhibition of signal transduction. mTOR inhibitors sirolimus and everolimus are well known drugs and were initially used as immunosuppressive therapy in the field of solid organ transplantation. Other mTOR inhibitors are deforolimus (an investigational agent in the management of sarcoma and breast cancer), zotarolimus (used as a coating in stents), ridaforolimus (used in eluting stents and in advanced malignancies) and temsirolimus (used in kidney cancer and under investigation for lymphoma, breast cancer, and other tumours). The use of mTOR inhibitors in the field of haematology is very broad, varying from standardised first-line treatments (GvHD), to promising new drug regimens whose results are waiting to be confirmed in multicentre studies (autoimmune haemolytic anaemia, myelodysplasias, acute leukaemias, lymphomas, multiple myeloma, Waldenstrom macroglobulinemia, immune thrombocytopenic purpura, acquired aplastic anaemia, pure red cell aplasia, and autoimmune lymphoproliferative syndrome). mTOR inhibitors have been used in acute leukaemias in combination strategies with other chemotherapies and could be an effective treatment for patients with acute high-risk leukaemias, after an accurate stratification of patients [67]. Platzbecker et al. treated 19 patients with myelodysplastic syndrome with sirolimus as a single agent and demonstrated a good efficacy of remission in patients with advanced disease, although this efficacy was not seen for low-risk myelodysplastic syndrome [68]. The combination of JAK inhibitors such as ruxolitinib with everolimus has showed some synergy in inducing a blockade of proliferation, which is promising for the treatment of myeloproliferative neoplasms [69]. The American Society of Hematology recommended sirolimus as a second-line treatment of autoimmune haemolytic anaemia in the case of failure to steroid treatment [70]. However, the primary use of mTOR inhibitors rests in the prophylaxis of GvHD in patients who have undergone allogenic transplantation. Sirolimus has shown clinical benefit in both prevention and treatment of GvHD. In 2008, Armand et al. demonstrated that sirolimus combined with tacrolimus prevented GvHD in lymphoma patients after bone marrow transplantation [71]. In a recent study, after marrow transplant, the standard prophylaxis regime for GvHD (cyclosporine and mycophenolate mofetil) was compared to a combination of three drugs (cyclosporine, mycophenolate mofetil, and sirolimus). Consistent with previous findings, the incidence of grade II–IV GvHD was lower in the group treated with three drugs (26%) than in the standard group (52%) after 100 days [72].

A common side effect of mTOR inhibitors is metabolic toxicity. in observational studies, an incidence of mTOR-inhibitor-related hyperlipidaemia of up to 75% has been reported [73]. However, most patients experience a mild dyslipidaemia; only a few patients have moderate or severe increases in triglycerides or cholesterol concentrations. Everolimus (1.5 mg/day) is associated with significantly lower concentrations of triglycerides and LDL after 6 months of therapy compared with sirolimus (3.0 mg/day). A review on the incidence of adverse lipid-related events in transplantation studies of everolimus or sirolimus demonstrated hypertriglyceridaemia in 4% of patients taking everolimus (with reduced- or standard-dose tacrolimus or cyclosporine) and in 21 to 57% of patients taking sirolimus (with cyclosporine ± corticosteroids); the incidence of hypercholesterolemia was, respectively, 16% and 20 to 46% [70]. In a meta-analysis of patients with advanced malignancies treated with mTOR inhibitors (everolimus, temsirolimus, and ridaforolimus) the incidence rate of hypertriglyceridaemia was 35% (severe hypertriglyceridaemia 3%) and the incidence rate of hypercholesterolemia was 32% (severe hypercholesterolemia 3%) [74]. mTOR inhibitors (sirolimus and everolimus) have a dose-dependent lipid-increasing effect, that is greater compared to CNIs. Increased levels of TC, LDL-C, and TG have been reported with mTOR inhibitor therapy, which are greater than with cyclosporine monotherapy [75]. The increase in the fraction of sd-LDL turns out to be proportional to the time elapsed since transplantation [52]. In effect, mTOR inhibitors worsen cyclosporine-induced hypercholesterolaemia and steroid-induced hypertriglyceridaemia. Sirolimus works by reducing the activity of 27-hydroxylase and inhibits the transcription of the gene for the LDL receptor in hepatic cells, which results in a reduced clearance of LDL [76]. Chronic rapamycin treatment in rats induces glucose and insulin intolerance and downregulates genes implicated in lipid uptake and storage in adipose tissue [77]. Other mechanisms suggested for the pathogenesis of mTOR-inhibitor-induced hypertriglyceridaemia are: (1) an upregulation of the gene CIII A, which is an inhibitor of LPL with a consequent inhibition of TG and VLDL catabolism leading to hypertriglyceridaemia [60]; (2) an upregulation of adipocyte fatty-acid-binding protein expression, which may contribute to hypertriglyceridaemia [78]; and (3) a reduced catabolism of lipoproteins containing apo B 100 leading to an elevation in VLDL concentrations [79]. In patients with mTOR-inhibitor-related severe hypertriglyceridaemia, fenofibrate should be considered the drug of first choice; if LDL-C concentrations are elevated too, a statin therapy could be added under strict surveillance for potential drug-related myopathy [80]. Despite optimal therapy for dyslipidaemia, many patients (up to 50%) with hyperlipidaemia after mTOR inhibitors therapy do not achieve the cholesterol and triglyceride concentrations recommended by the most prestigious scientific societies. However, the treatment of transplant patients with drug-induced dyslipidaemia was addressed in a recent review [81]. The other side of the coin is the protective role played by some mTOR inhibitors against atherosclerosis and thrombosis. Several pieces of evidence of the antithrombotic effects of sirolimus and everolimus have been reported. In a recent review, Arachchillage and Laffan demonstrated that these drugs, by inhibiting the mTOR complex, prevent endothelial proliferation and intimal hyperplasia in patients affected by antiphospholipid syndrome (APS). In patients with APS, endothelial proliferation and intimal hyperplasia lead to microthrombosis, which is one of the main pathological manifestations of APS that occur alone or in combination with large-vessel thrombosis [82]. Moreover, everolimus has been shown to have antithrombotic effects, since it improves microcirculatory derangements in postischemic pancreatitis experimentally, through the modulation of the expression of several inflammatory proteins such as interleukin 6, vascular endothelial growth factor (VEGF) and toll-like receptor 4 [83]. Wenzel at al. suggested that the early use of mTOR inhibitors may limit the inflammatory increase of IL 6 and VEGF after ischaemia reperfusion injury [84].

## 7. All-Trans Retinoic Acid

Acute promyelocytic leukaemia (APL) is the M3 subtype of acute myeloid leukaemia, characterised by a mutation of the chimeric oncoprotein PML-RARα. Even though a raised body mass index (BMI) and high prevalence of obesity are reported in patients with APL, this does not seem to be associated with abnormalities of the lipid profile at the time of diagnosis [85]. Retinoid hyperlipidaemia has been recognised since the mid-1970s when synthetic retinoids became available for clinical use, especially in dermatology [86]. All-trans retinoic acid (ATRA) and the trioxide of arsenic (As2O3) have long been used successfully in treating APL. However, in recent years, a lot of attention has been paid towards hypertriglyceridaemia induced by ATRA when given for APL. Two mechanisms have been suggested to explain this. Firstly, the action of ATRA increases lipid synthesis in the liver, increasing the serum levels of cholesterol and triglycerides. Apo CIII production and plasma concentrations are increased in humans on retinoid treatment which suggests that this effect, at least in part, could participate in the frequently observed hyper-triglyceridemic effects of ATRA [87]. Secondly, metabolites, comprised of cytokines and adipokines, produced by APL cells could contribute to ATRA-induced hypertriglyceridaemia [88]. Most of the time, ATRA induces a modest hyperlipidaemia controllable with a correct dietary approach, but sometimes, cases of severe hypertriglyceridaemia (rarely with pancreatitis) have been reported, requiring a suspension of the drug. In a landmark study on the use of ATRA for the treatment of newly diagnosed APL, 18% of the patients developed hypercholesterolemia (9% mild, 9% moderate) and 50% hypertriglyceridaemia (17% mild, 29% moderate, and 4% severe) [89]. On the other hand, one must consider that ATRA improves atherosclerosis in Apo-E mice and that ATRA improves insulin sensitivity and increases lipid catabolism by activating retinoic acid receptor (RAR) and peroxisome proliferator-activated receptor (PPAR) β/δ in obese mice [90,91].

## 8. Corticosteroids

Here are just a few words on corticosteroids due to their use as a single agent or the fact they are added to other drugs in many haematological diseases.

It has been postulated that the chronic use of glucocorticoids can cause secondary dyslipidaemia, but the severity of hyperlipidaemia in various clinical conditions is extremely variable and previous studies have given conflicting results and have been incoherent [92]. Observational studies looking at the use of steroids in the treatment of asthma, rheumatoid arthritis, or connective tissue diseases have shown increased serum levels of triglycerides, LDL, and total cholesterol [93]. These diseases are all conditions that require long term use of steroids. One study demonstrated that premenopausal women who took steroids for an average of 3.1 years developed significantly raised total cholesterol and reduced levels of HDL-C. In contrast, a study looking at female patients with asthma found significantly increased levels of serum TG but no changes in cholesterol levels [94].

Exposure to systemic steroid therapy, used as a fundamental part of therapeutic protocols for the treatment of ALL in children, is associated with an increased risk of developing hypertriglyceridaemia, metabolic syndrome, obesity, and hypertension in adult life [95].

Potential mechanisms explaining the effects of steroids on the lipid profile are multifactorial. Glucocorticoids induce hyperinsulinemia and hepatic insulin resistance. They stimulate hepatic gluconeogenesis and increase VLDL synthesis contributing to hypertriglyceridaemia. Moreover, steroid-induced hyperinsulinemia stimulates an increased rate of VLDL production and augmented plasma VLDL concentrations [96,97].

## 9. Anti-CD20

Rituximab is a chimeric monoclonal antibody specific to CD20, an antigen expressed on B-cells. The efficacy of rituximab for the treatment of non-Hodgkin B-cell lymphoma and its relative lack of toxicity have brought its use into the majority of treatment protocols for B-cell lymphoma, as well as a broad spectrum of other B-cell disorders, including autoimmune diseases and other malignancies. Treatment with rituximab potently suppresses systemic inflammation, improves the lipid profile and the atherogenic index, and appears to decrease carotid intima–media thickness [98,99].

However, we must highlight that two subtypes of B-lymphocytes exist, which have contrasting effects on atherosclerosis: B1 cells are protective against atherosclerosis whereas B2 cells are proatherogenic. Treatment with rituximab suppresses the atherogenic effect whilst preserving the atheroprotective aspects of B1 cells, with the consequent inactivation of T-cells and macrophages, which decreases the production of proinflammatory cytokines and antibodies and ultimately the oxidation of LDL. The selective inhibition of T-cells could also indirectly influence endothelial function and prevent major vascular events. A recent study has demonstrated that a single infusion of rituximab effectively suppresses circulating mature B-cells after just 30 min following infusion of doses up to 200 mg, suggesting the feasibility of a “fire and forget” approach, with a rapid modulation of immune responses during the first critical months following myocardial infarction [100]. Moreover, rituximab was used in a case of severe acquired autoimmune hypertriglyceridaemia resistant to traditional triglyceride-lowering therapies and it was observed to reduce plasma anti-LPL antibody levels and resulted in an improvement of hypertriglyceridaemia [101].

Statin therapy in patients with diffuse large B-cell lymphoma under treatment with rituximab plus a regimen of cyclophosphamide, doxorubicin, vincristine, and prednisolone (CHOP) is still debated. Some authors claim that the depletion of cholesterol induced by statins could determine conformation change in CD20, which could impair the binding of rituximab to CD20 preventing its killing effect on lymphoma cells [102]. Other studies suggest that statins could be a therapeutic strategy to ameliorate responses to rituximab plus CHOP in patients with diffuse large B-cell lymphoma [103].

In conclusion, unlike the other haematological drugs discussed so far, which in most cases have a negative effect on the lipid profile and atherosclerosis, rituximab appears to have a favourable effect on lipid balance and ultrasound proxies of atherosclerosis.

## 10. Conclusions

The present review summarises the principal actions on atherosclerosis and lipid metabolism of drugs used in the treatment of haematological diseases. Table 2 gives a brief overview of these pharmacological side effects. However, the magnitude of the effect on lipid metabolism is quite different among the different haematologic drugs examined in this review. Although it is not always clear if these effects on lipid metabolism have an impact on CV outcomes, awareness that drug-induced dyslipidaemia occurs and allowing a cost-benefit analysis for each individual patient is essential for a “global” treatment of haematological disorders. 

## Figures and Tables

**Figure 1 biomedicines-10-01935-f001:**
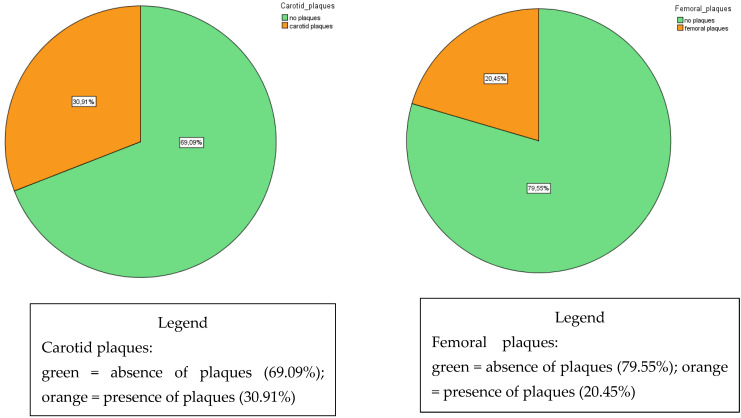
Prevalence of carotid and femoral plaques prior to initiating second- and third-generation tyrosine kinase inhibitors (nilotinib and ponatinib; *n* = 55).

**Table 1 biomedicines-10-01935-t001:** Cardiovascular baseline data for 35 patients undergoing treatment with nilotinib.

	Mean ± SD	Range
Age (years)	**51.1 ± 13.6**	**25–79**
Sex (M/F)	**51%**	
Systolic arterial blood pressure (mmHg)	**129 ± 18**	**106–180**
Diastolic arterial blood pressure (mmHg)	**82 ± 6**	**70–90**
Ankle–brachial pressure index (Right)	**1.14 ± 0.12**	**0.92–1.42**
Ankle–brachial pressure index (Left)	**1.14 ± 0.17**	**0.67–1.54**
Carotid intima–media thickness (right) (mm)	**1.36 ± 0.7**	**0.6–3.0**
Carotid intima–media thickness (left) (mm)	**1.23 ± 0.7**	**0.7–4.0**
Cardiac ejection fraction (%)	**60.7 ± 3.5**	**55–65**
Smokers (%)	**60% No** **27% Ex** **13% Yes**	
Hypertensive (%)	**29%**	
Diabetic (%)	**12%**	

**Table 2 biomedicines-10-01935-t002:** Principal drugs used in haematology which cause dyslipidaemia or vasculopathy.

Name of the Drug	Haematological Diseases in Which They Are Used	Mechanism of Action	Main Effects on Lipids or Vasculopathy	Pathogenesis	Treatment (Lifestyle Modifications Are Always Encouraged)
Nilotinib	LMC	Second-generation tyrosine kinase inhibitor	↑ LDL-C [23] ↑↑ Vascular disease (PAD) [17,18,19,20]	Not fully understood prothrombotic and antiangiogenic effects	-Statins -Vasoactive drugs -Nilotinib discontinuation
Ponatinib	LMC	Third-generation tyrosine kinase inhibitor	↑ Vascular disease (PAD) [27]	Not fully understood prothrombotic effects	-Statins -Vasoactive drugs -Ponatinib discontinuation
Ruxolitinib	Idiopathic myelofibrosis	JAK1/2 inhibitor	↑↑ TG (together with sirolimus) [35,36]	Dysregulation of the leptin receptor [37]	In severe hyperTG: -Ruxolitinib discontinuation -Plasma exchange
PEG-asparaginase	LLA	Depletion of amino acid L-asparagine	↑↑ TG (together with corticosteroids) [44,45]	↑ VLDL [39] ↓ LPL [40]	-Omega 3 -Fibrates In severe hyperTG: -PEG-ASP discontinuation -Plasma exchange
Cyclosporine	Prophylaxis of GvHD after haematopoietic stem-cell transplantation, Castleman disease	Calcineurin inhibitor	↑ LDL [51] ↑TG [51] ↑ sdLDL [60]	↑ Insulin resistance [54] ↑ Cholesterol synthesis [55] ↓ Clearance VLDL [57] ↑ Apo CIII [60] ↓ LPL [58,59]	-Fluvastatin, pravastatin -fibrates
Tacrolimus	Prophylaxis of GvHD after haematopoietic stem-cell transplantation	Calcineurin inhibitor	↑ LDL [104] ↑ TG [104]	↑ Insulin resistance [61] ↓ Akt phosphorylation [63] ↓ Circ-RNA [62]	-Fluvastatin, pravastatin -fibrates
Sirolimus	Prophylaxis of GvHD after haematopoietic stem-cell transplantation; myelodysplastic syndrome; autoimmune haemolytic anaemia	mTOR inhibitor	↑ TG [60,73,74] ↑ VLDL [79] ↑ sd LDL [52] ↑ LDL [73,74]	↑ Apo CIII [60] ↑ Fatty acid binding protein [78] ↑ Gluconeogenesis [77] ↓ LPL activity [75] ↓ Clearance LDL [75,76]	-Fenofibrate -Statins (↑ risk of rhabdomyolysis) -Omega-3-acid ethyl esters
Everolimus	Prophylaxis of GvHD after haematopoietic stem-cell transplantation	mTOR inhibitor	↑ TG [73,75] ↑ VLDL [80] ↑ sd LDL [52] ↑ LDL [75]	↑ Apo CIII [80] ↓ LPL activity [75,80] ↓ Clearance LDL [75,80]	-Fenofibrate -Statins (↑ risk of rhabdomyolysis) -Omega-3-acid ethyl esters
ATRA	APL	Antineoplastic agent	↑↑ TG [89]	↑ Hepatic TG synthesis [105] ↑ apo CIII [87]	-Omega-3-acid ethyl esters -Fibrates -ATRA withdrawal
Corticosteroids	Used in conjunction with other agents in multiple haematological diseases	Anti-inflammatory steroid	↑ TG [95] ↑= LDL [93,94]	↑ Insulin resistance [96] ↑ VLDL [97]	-Statins -Fibrates
Rituximab	Non-Hodgkin lymphomas	Anti-CD20	↓ Carotid IMT [98,99] ↓ TG [101]	Suppression of active B2 cells [106]	Not necessary

Legend: ↑ increase; ↑↑ marked increase; ↓ decrease.

## Data Availability

Data on the prevalence of carotid and femoral plaques are available upon request at: Department of Medicine and Medical Specialties, A. Cardarelli Hospital. E-mail arcangelo.iannuzzi@aocardarelli.it.

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
