# Peer review of "Haematological Drugs Affecting Lipid Metabolism and Vascular Health"

_biomedicines, 2022, doi:10.3390/biomedicines10081935_

Round 1

Reviewer 1 Report

If it is possible thanks to the literature to have a value of triglycerides and cholesterol of the different drugs. I would also like to understand roughly what order measurement of lipid metabolism among the different drugs.

Author Response

If it is possible thanks to the literature to have a value of triglycerides and cholesterol of the different drugs. I would also like to understand roughly what order measurement of lipid metabolism among the different drugs.

We added a paragraph to each chapter of the manuscript to explain the magnitude of action of the different drugs on cholesterol and/or triglycerides concencentrations.

Reviewer 2 Report

Parrella et al. prepared a comprehensive review of the side effects of the main drugs used in haematology, the mechanisms of action of those side effects, and effective therapies for those side effects. The English language usage is clear and professional, and the manuscript is overall easy to follow. This review will highlight the importance of paying attention to the side effects of those drugs and will be of help to diagnose drug-induced hyperlipidaemia early to avoid severe results. I therefore recommend it to be published in Biomedicines, after considering the following suggestions about the figures and tables:

(1)           The keywords should better represent the topics of the manuscript and make this manuscript more searchable. For example, “drug” is too generic; “side effects” could be added as a keyword.

(2)           The font size of Figure 1 should be larger, otherwise it is hard to read.

(3)           Please include in Table 2 caption or footnote what those up arrows or down arrows mean; Do they mean “up-regulated” and “down-regulated”? Why sometimes double arrows were used, and sometimes single arrows were used? Why different styles of up arrows were used?

(4)           Table 2 should be better formatted, currently it is hard to read: some word is divided into two parts.

(5)                It would be nice to provide references in table 2 for each drug.

Author Response

Parrella et al. prepared a comprehensive review of the side effects of the main drugs used in haematology, the mechanisms of action of those side effects, and effective therapies for those side effects. The English language usage is clear and professional, and the manuscript is overall easy to follow. This review will highlight the importance of paying attention to the side effects of those drugs and will be of help to diagnose drug-induced hyperlipidaemia early to avoid severe results. I therefore recommend it to be published in Biomedicines, after considering the following suggestions about the figures and tables:

(1)           The keywords should better represent the topics of the manuscript and make this manuscript more searchable. For example, “drug” is too generic; “side effects” could be added as a keyword.

We changed the keywords, following the advice of the reviewer: Haematological drugs ; side effects;  lipid metabolism; hypertriglyceridemia ; hypercholesterolemia ;  atherosclerosis ;  vasculopathy.

(2)           The font size of Figure 1 should be larger, otherwise it is hard to read.

 We added a legend, explanatory of the findings.

(3)           Please include in Table 2 caption or footnote what those up arrows or down arrows mean; Do they mean “up-regulated” and “down-regulated”? Why sometimes double arrows were used, and sometimes single arrows were used? Why different styles of up arrows were used?

We added a Legend: ↑ increase; ↑↑ marked increase; ↓ decrease. In the current table 2 all the arrows are of the same style.

(4)           Table 2 should be better formatted, currently it is hard to read: some word is divided into two parts.

We tried to improve the readability of the Table 2.

(5)                It would be nice to provide references in table 2 for each drug.

We followed the advice of the reviewer.